# A tRNA half modulates translation as stress response in *Trypanosoma brucei*

Roger Fricker[1,2], Rebecca Brogli[1,2], Hannes Luidalepp[1], Leander Wyss[1,2], Michel Fasnacht[1,2], Oliver Joss[1], Marek Zywicki[3], Mark Helm [4], André Schneider[1], Marina Cristodero[1] & Norbert Polacek[1]

In the absence of extensive transcription control mechanisms the pathogenic parasite *Trypanosoma brucei* crucially depends on translation regulation to orchestrate gene expression. However, molecular insight into regulating protein biosynthesis is sparse. Here we analyze the small non-coding RNA (ncRNA) interactome of ribosomes in *T. brucei* during different growth conditions and life stages. Ribosome-associated ncRNAs have recently been recognized as unprecedented regulators of ribosome functions. Our data show that the tRNA$^{Thr}$ 3´half is produced during nutrient deprivation and becomes one of the most abundant tRNA-derived RNA fragments (tdRs). tRNA$^{Thr}$ halves associate with ribosomes and polysomes and stimulate translation by facilitating mRNA loading during stress recovery once starvation conditions ceased. Blocking or depleting the endogenous tRNA$^{Thr}$ halves mitigates this stimulatory effect both in vivo and in vitro. *T. brucei* and its close relatives lack the well-described mammalian enzymes for tRNA half processing, thus hinting at a unique tdR biogenesis in these parasites.

[1] Department of Chemistry and Biochemistry, University of Bern, Freiestrasse 3, 3012 Bern, Switzerland. [2] Graduate School for Cellular and Biomedical Sciences, University of Bern, 3012 Bern, Switzerland. [3] Department of Computational Biology, Institute of Molecular Biology and Biotechnology, Adam Mickiewicz University, Umultowska 89, 61-614 Poznan, Poland. [4] Institute of Pharmacy and Biochemistry, Johannes Gutenberg-University of Mainz, Staudingerweg 5, D-55128 Mainz, Germany. These authors contributed equally: Roger Fricker, Rebecca Brogli. Correspondence and requests for materials should be addressed to M.C. (email: marina.cristodero@dcb.unibe.ch) or to N.P. (email: norbert.polacek@dcb.unibe.ch)

The pathogenic protozoan parasite *Trypanosoma brucei* and its relatives are causative agents of human African trypanosomiasis in sub-Saharan regions and other devastating diseases that are difficult to treat. *T. brucei* has a complex life cycle involving two different hosts, an insect and a mammal, and possesses clearly distinguishable developmental stages[1]. Despite the fact that this protozoan organism has to adapt to different environments it largely lacks the ability to regulate transcription of protein coding genes. Therefore, and in contrast to other eukaryotes, they heavily rely on posttranscriptional means to regulate gene expression[2]. At first sight, the lack of transcriptional control seems to be energetically wasteful, but it may help the trypanosomes to react faster to environmental challenges and can even be beneficial during host transmission[3]. While well understood posttranscriptional mechanisms of gene regulation such as the RNAi pathways are at work in most eukaryotes, miRNA-guided translation regulation seems to be absent in trypanosomes[4]. Even though translation control is pivotal for the regulation of gene expression in *T. brucei*[5] there is little information about how these parasites regulate protein biosynthesis at the molecular level. Recently we have identified ribosome-associated ncRNAs (rancRNAs) in the archaeon *H. volcanii*[6,7] and in *Saccharomyces cerevisiae*[8] as potent agents involved in translation control mechanisms. RancRNAs represent an emerging class of translation regulators acting primarily during stress response and include small as well as long ncRNAs (reviewed in ref. [9]). Over the past years rancRNAs have been identified as riboregulators in all domains of life. The main advantage of rancRNA-mediated translation control is its immediate availability, since the regulatory entity is a small RNA typically derived from already existing RNA species and capable of targeting the ribosome as the main component of the translation machinery.

In order to investigate if rancRNAs play crucial roles in *T. brucei* we have analyzed the small ncRNA interactome of ribosomes isolated from two different developmental stages of the parasites exposed to different environmental conditions. We reveal tRNA halves as one of the most strongly affected classes of transcripts upregulated during nutrient deprivation and stationary phase. In particular, the abundance of the tRNA^Thr 3′ half is significantly increased and it was found to interact with ribosomes and polysomes upon starvation. This particular tRNA half-ribosome interaction stimulates protein biosynthesis in vitro as well as in vivo and appears to play a role primarily during the stress recovery phase of *T. brucei*.

## Results

**Ribosome-associated ncRNAs in *T. brucei*.** To gain insight into the composition and putative biological function of the emerging class of rancRNAs[9], we generated a cDNA library encoding small RNAs in the size range between 20 and 300 nucleotides that co-purify with cytosolic ribosomes in *T. brucei*. To this end ribosomes and polysomes from the procyclic as well as from the bloodstream forms of *T. brucei* cells in the exponential and stationary phases were collected. In addition, ribosomes were isolated for rancRNA identification from procyclic cells after heat shock, cold shock or nutritional stress. Subsequent to deep sequencing analysis the obtained reads were analyzed using a modified version of the previously established APART pipeline[10]. After trimming the adaptor sequences and quality control 30.1 million reads remained (in average 5 million reads per cDNA library) that were further analyzed and resulted in 3596 putative rancRNA candidates. All sequencing reads can be accessed via the European Nucleotide Archive number PRJEB24915. The majority of the sequencing reads mapped to ribosomal RNA loci, which is not surprising given the fact that the 28S rRNA of *T. brucei*

ribosomes is naturally split into six fragments, most of them in the size range under investigation. The second largest pool of sequences mapped to tRNA loci (Fig. 1a). By comparing the distribution of mapped reads between the different growth or stress conditions, it was evident that upon nutrient deprivation and during the stationary phase tRNA-derived reads became a very abundant species and dominated the sequenced rancRNA pool (Fig. 1a). Detailed information on all tRNA-derived sequencing reads are compiled in Supplementary Data 1. Length distribution of tRNA-derived reads revealed a clear peak at around 33 nucleotides, which primarily represents tRNA halves (Fig. 1b). By analyzing the abundance levels of tRNA halves across different growth conditions a significantly high correlation became evident between samples originating from heat shock, cold shock and exponentially growing *T. brucei* samples on the one side and starved and stationary cells on the other (Fig. 1c). Furthermore this analyses revealed tRNA-fragment abundance levels isolated from bloodstream form samples to be very distinct from all others and indicate a unique expression pattern, including a distinct fraction of 21 nucleotides long tRNA fragments (Fig. 1b, c). tRNAs carry multiple post-transcriptional modifications, which can hamper reverse transcription during cDNA library preparation and thus could potentially skew conclusions about tRNA fragment lengths and cellular abundance. In order to circumvent this potential limitation and to validate the rancRNA-seq data we performed comprehensive tRNA halves northern blot analyses on total RNA isolated from *T. brucei* exposed to different stress conditions. The results support the sequencing data by showing different levels of tRNA halves for almost all tRNA isoacceptors (Supplementary Figure 1). These experiments reveal that most of the detectable tRNA halves originate from the 5′ part of tRNAs and thus further confirm the sequencing data. In these northern analyses we noticed tRNA processing into halves as a consequence of different growth and stress condition to be tRNA species-dependent and thus quite heterogeneous (Supplementary Figure 1).

**The abundance of tRNA halves varies during stress.** To test which of these tRNA halves also associate with ribosomes, we repeated the northern blot analyses on the crude ribosome pellet. Since the sequencing and northern blot data indicated most abundant tRNA processing into halve-mers under nutrient deprivation (after incubating cells in PBS or during stationary phase), we compared ribosomes isolated from starved and exponentially growing *T. brucei* cultures (Fig. 1d). At least ten tRNA halves could be clearly detected on these northern blots. Among these potentially ribosome-associated tRNA fragments the tRNA^Thr 3′ half as well as the two 5′ halves originating from tRNA^Ala and tRNA^Asp were the most abundant ones (Fig. 1d) and showed differential cellular expression during the growth conditions tested (Supplementary Figure 1). We first focused on the 3′ tRNA half deriving from the tRNA^Thr isoacceptor harboring the AGU anticodon (Tb427_10_tRNA_Thr_1). Based on the sequencing reads, this half is particularly abundant during nutrient deprivation and in stationary phase (Supplementary Data 1, Supplementary Figure 2a). Northern blot analysis confirmed this abundance pattern and demonstrated that the 3′ tRNA^Thr half accumulates in a time dependent manner in procyclic *T. brucei* cells during starvation and in late stationary phase (Fig. 2a, b). The time point of the first tRNA^Thr 3′ half detection (two hours of nutrient deprivation) coincides with moving impairment of *T. brucei* cells. Placing the starved cells back into normal growth media results in resumed flagellum movement after 30 min whereas the tRNA^Thr half remained at constant levels during this recovery phase for up to 2 h (Fig. 2c).

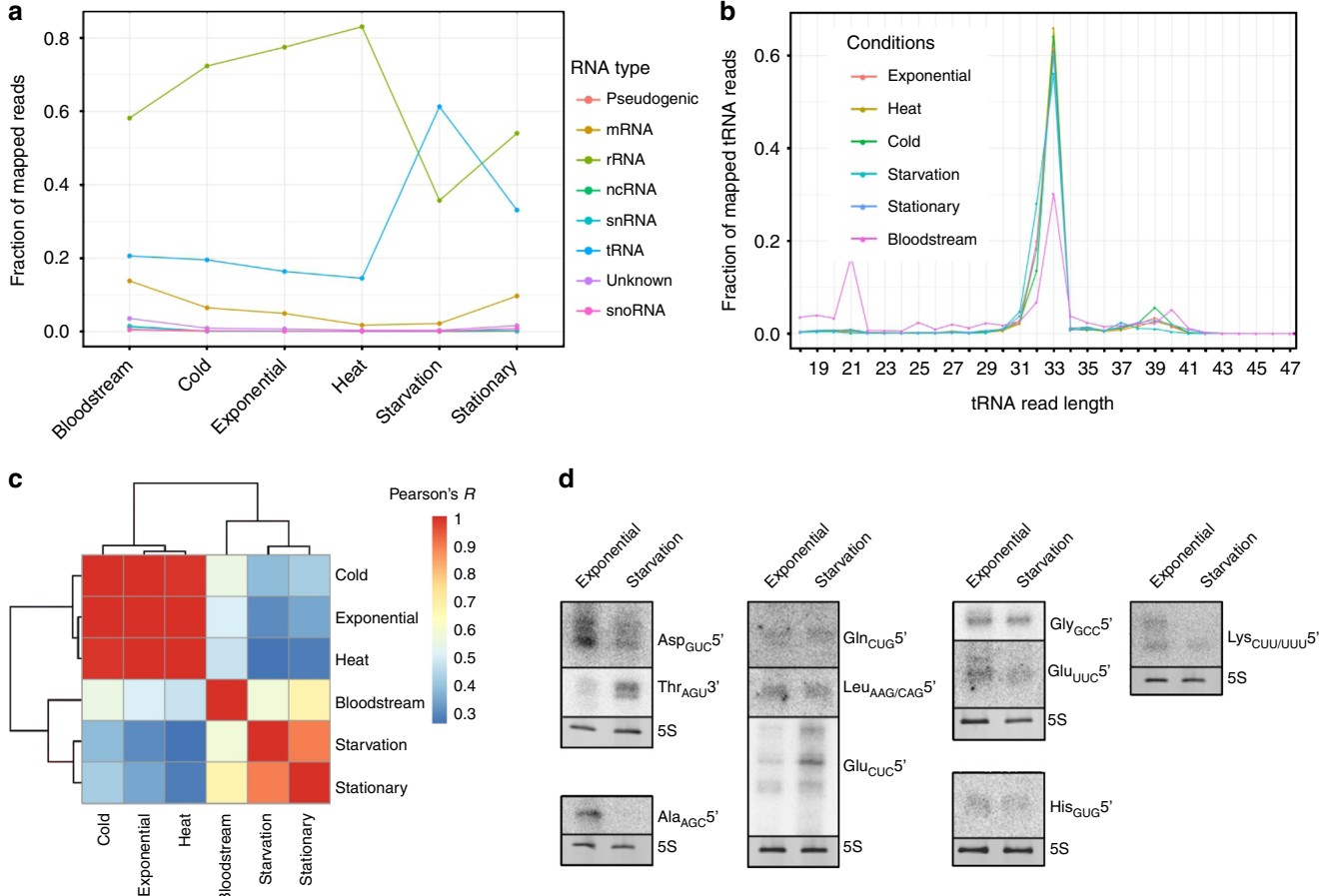

**Fig. 1** Profiling the abundance levels of rancRNA sequencing reads. **a** The distribution of sequencing reads fractions assigned to different RNA classes among different growth conditions and life stages of *T. brucei*. Note that the category 'ncRNA' includes annotated *T. brucei* ncRNAs not listed otherwise. **b** Read length distribution of tRNA-derived sequencing reads observed during different growth conditions. **c** Sample correlation matrix showing Pearson's correlation of expression levels of identified tRNA processing products between different growth conditions and life stages. **d** Ribosome-association of tRNA halves was assessed via northern blot analyses on RNA isolated from the crude ribosomal pellet from exponentially growing cells or from cells starved for two hours by incubating the parasite in PBS. The 5S rRNA (5S) served as loading control. tRNA isoacceptor anticodons and the origin of the tRNA halve (5′ or 3′) are indicated

In northern blots we noticed a slower migrating band above the mature tRNA$^{Thr}$ in starved cells that rapidly disappears during the recovery phase (Fig. 2). Against intuition, this slower migrating RNA band corresponds to the tRNA$^{Thr}$ lacking the 3′ CCA end (Supplementary Figure 3a, b). It is of note that almost all sequencing reads originating from tRNA$^{Thr}$ also lack the 3′ CCA ends during nutritional stress (Supplementary Figure 2b). This 3′ tRNA trimming is in fact a widespread phenomenon under nutrient deprivation in *T. brucei* (Supplementary Figure 1a) and thus potentially analogous to oxidatively stressed human tRNAs[11]. Under the starvation conditions applied (PBS incubation for at least 2 h) new tRNA transcription is likely negligible suggesting that tRNA halves are produced during nutritional stress from mature tRNAs lacking the 3′ CCA.

A different tRNA$^{Thr}$ half abundance pattern was observed in the bloodstream form of *T. brucei*. In contrast to the procyclic stage, which is the most prevalent form in the insect host, the tRNA$^{Thr}$ 3′ half is already present during exponential growth in the bloodstream form of the parasite. The tRNA half level in the bloodstream form remained constant during starvation and during the recovery phase (Fig. 2d).

Compared to the 3′ tRNA$^{Thr}$ half, a completely different expression pattern was evident for the tRNA$^{Ala}$ 5′ half (*Tb427.07.6821*) (Supplementary Figure 1). It is present in easily

detectable amounts during exponential growth in the procyclic as well as in the bloodstream form, while it is undetectable during starvation conditions in both forms of the parasite (Fig. 3a, b). Alleviating the nutrient deprivation by transferring the cells back into rich media resulted in the fast reappearance of the tRNA$^{Ala}$ half within 15 to 30 min. Also during the stationary growth phase this 5′ tRNA half declines in abundance as a function of increased cell density (Fig. 3c). This indicates that fragment accumulation is tRNA-specific and again further suggests that tRNA halves are not unspecific degradation intermediates.

**tRNA halves associate with ribosomes and polysomes.** Since we discovered the tRNA halves in an RNA-seq library designed to identify rancRNAs we next investigated whether the identified tRNA halves have affinity for ribosomal particles. Therefore polysome profiles of exponentially growing procyclic cells and cells starved for two hours were recorded in linear sucrose gradients. During starvation conditions the polysomes were markedly reduced and monosomes and ribosomal subunit peaks were increased compared to the exponential growing control (Fig. 4a). This indicates reduced translational rates and dimmed overall metabolic activities under growth conditions where nutrients are limiting. Fractions of the polysome gradients were collected, the

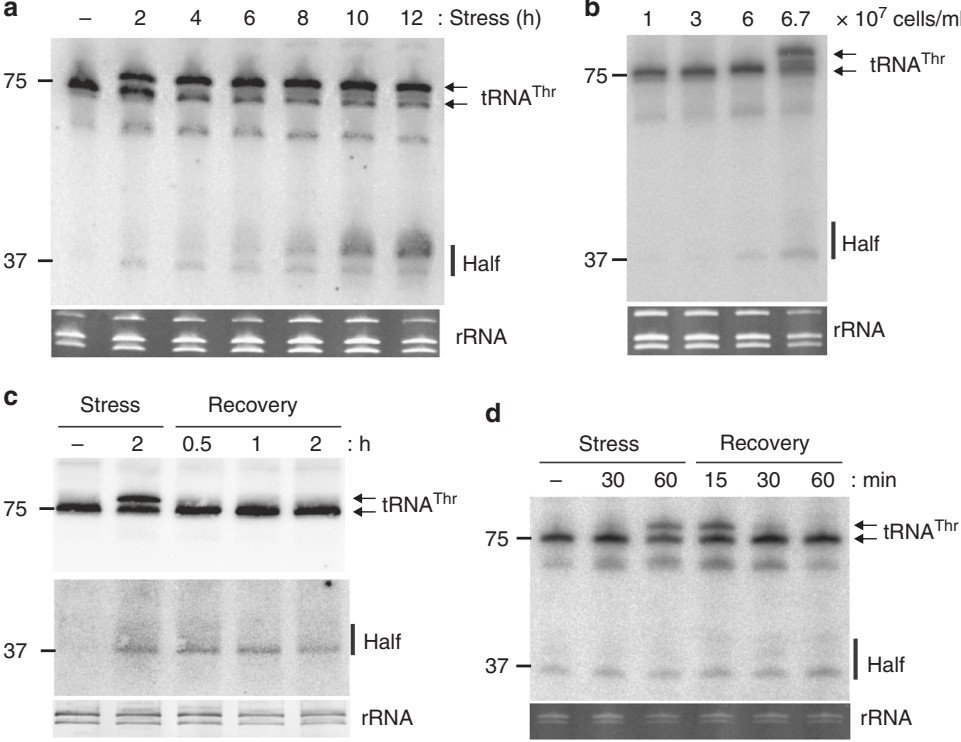

**Fig. 2** tRNA[Thr] 3′ half accumulates during stress in procyclic and bloodstream *T. brucei*. **a** *T. brucei* procyclic cells were starved for the indicated times by incubation in PBS. Total RNA was extracted and the presence of the tRNA[Thr] 3′ half (37 nucleotides long; see also Supplementary Figure 10) monitored by northern blot analysis. **b** The presence of the tRNA[Thr] 3′ half was investigated as in **a** in procyclic cells grown to different cell densities (indicated on top of the panel). **c** Same as **a** but cells were allowed to recover in normal media for the indicated periods of time after nutritional stress. In the lower panel the contrast of this part of the blot was adjusted to more clearly see the tRNA[Thr] 3′ half. **d** *T. brucei* bloodstream cells were stressed by incubation in PBS and then allowed to recover in normal growth media. The presence of the tRNA[Thr] 3′ half was analyzed as described in **a**. In all cases the EtBr-stained rRNAs serve as loading controls

RNA isolated and used for northern blotting. When probing for the tRNA[Thr] 3′ half after nutritional stress, signals were detected in the fractions corresponding to the large ribosomal subunit (60S), the monosomes (80S), the polysomes and in light fractions at the top of the gradient (free RNA) (Fig. 4b). This indicates that the tRNA[Thr] half indeed associates with ribosomes in vivo and its binding site apparently resides in the large ribosomal subunit. Even though the amount of polysomes in the sucrose gradient of starved cells is below the limit of the applied detection system (Fig. 4a), a clearly visible tRNA[Thr] half signal was detectable on northern blots (Fig. 4b). This indicates that a fraction of tRNA halves accumulates on the few polysomes that are present in starved cells. Quantification of the northern blot signals corresponding to the full-length tRNA[Thr] and the tRNA[Thr] 3′ half in the polysomal fraction demonstrated that 55% of the signal derives from the tRNA half-mer. In human cells it was reported that certain tRNA halves promote stress granule (SG) formation and even become integral to the stress granules[12,13]. SG formation is not only beneficial for survival during challenging conditions but also for recovery from stress[14]. Therefore, we were interested if the *T. brucei* tRNA[Thr] 3′ half modulates SG formation and/or turnover. For this purpose, we used a cell line expressing the YFP-tagged granule marker DHH1 (ref. [15]) and followed the assembly of SGs in the presence of the tRNA[Thr] 3′ half. After two hours of starvation stress granules were readily observed in the parasite (Supplementary Figure 4). However, stress granule formation does not appear to correlate with tRNA[Thr] halves abundance in *T. brucei*, since increasing its intracellular concentration by electroporation of 3′ half transcripts did not influence stress granule formation or stability

(Supplementary Figure 4). Furthermore, in the sucrose gradients stress granules do not co-sediment with the ribosome or polysome fractions under the applied conditions (Fig. 4c). These findings support the view that tRNA[Thr] 3′ halves (i) do not modulate genuine stress granule formation in *T. brucei* but (ii) bind to ribosomes and polysomes in vivo. To corroborate ribosome association we performed in vitro binding studies using gradient-purified *T. brucei* 80S ribosomes isolated either from exponentially growing or from starved cells. In these binding experiments ribosome association could be confirmed and furthermore a preferential interaction of the tRNA half with stressed ribosomes was observed (Fig. 4d).

**The tRNA[Thr] half stimulates *T. brucei* translation**. Having established a ribosome-association of the tRNA[Thr] 3′ half we next tested whether this interaction has a functional consequence on the ribosomes' performance. Therefore, we established an in vitro translation assay with *T. brucei* cell extracts using the total endogenous mRNA pool as template and [35]S-methionine incorporation into proteins as readout (Fig. 5a). In the presence of the ribosome-targeting antibiotic puromycin all radiolabeled bands were drastically reduced demonstrating that the [35]S-methionine labeling of proteins was translation dependent. When the assay was performed in the presence of in vitro transcribed tRNA[Thr] 3′ halves we observed a slight but reproducible stimulatory effect on translation of about 20% (Fig. 5a). However, tRNA[Thr] 3′ halves containing the CCA-tail (a species that was not detected in our bioinformatics analyses; Supplementary Figure 2b) were unable to stimulate translation. To investigate if the failure of translation

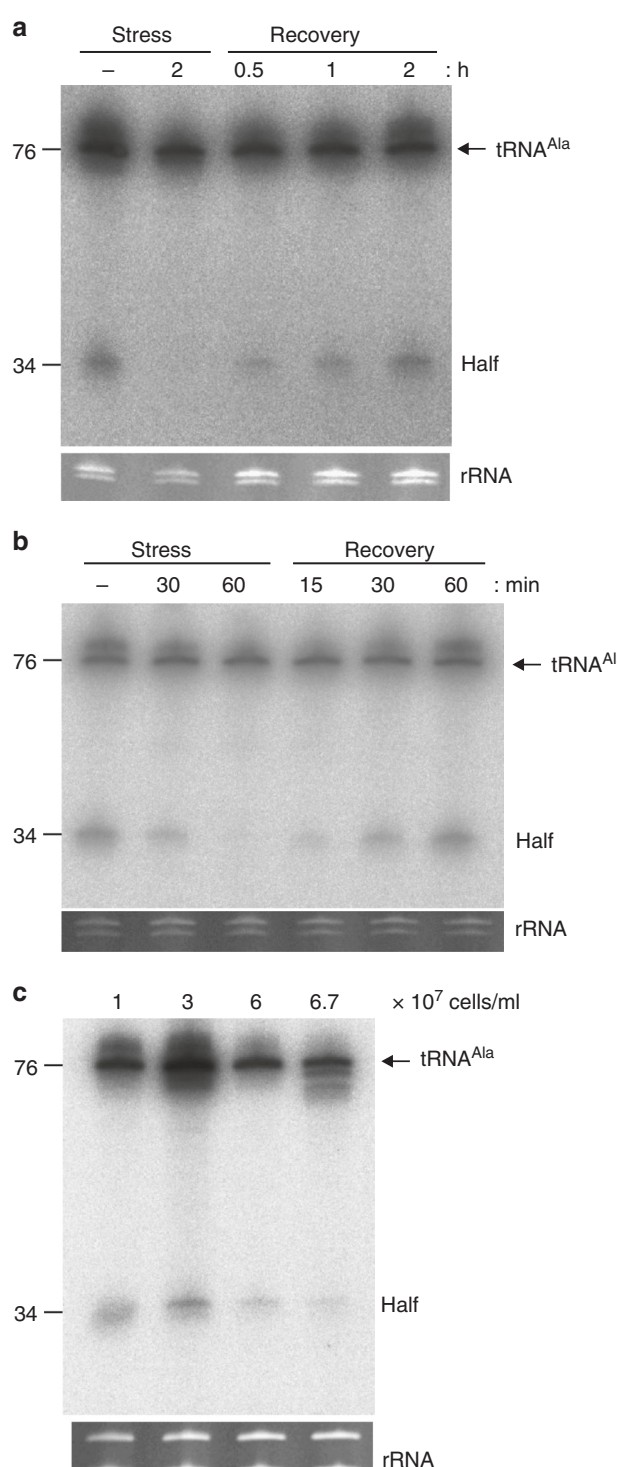

**Fig. 3** tRNA$^{Ala}$ 5′ half is present under normal growth but disappears upon stress in procyclic and bloodstream *T. brucei*. **a** *T. brucei* procyclic cells were starved for two hours by incubation in PBS. Subsequently, the cells were allowed to recover in normal media for the indicated periods of time. Total RNA was extracted and the presence of the tRNA$^{Ala}$ 5′ half (length: 34 nucleotides) monitored by northern blot analysis. **b** *T. brucei* bloodstream cells were stressed by incubation in PBS for 30 or 60 min and then allowed to recover in normal growth media. The presence of the tRNA$^{Ala}$ 5′ half was analyzed as described in **a**. **c** The presence of the tRNA$^{Ala}$ 5′ half was investigated as in **a** in procyclic cells grown to different cell densities (indicated on top of the panel). In all cases the EtBr-stained rRNAs serve as loading controls

To investigate whether these in vitro effects have a physiological significance in the parasite we used electroporation to introduce synthetic tRNA halves into *T. brucei* cells. Conditions for small RNAs electroporation were optimized by using an siRNA (siRNA-315) targeting α-tubulin[16], which was shown to cause a morphological phenotype (the so called FAT cells) due to the accumulation of multiple nuclei because of incomplete cytokinesis[17]. The phenotype was observed 18 h after electroporation and, under our optimized electroporation conditions, 71% of cells displayed the FAT phenotype (Supplementary Figure 6).

We next introduced in vitro transcribed tRNA$^{Thr}$ halves into *T. brucei* cells. Northern blot analysis performed two hours after electroporation demonstrated easily detectable amounts of the in vitro transcribed tRNA halves inside the cells (Supplementary Figure 7a). Omitting the electroporation step resulted in the complete loss of the northern blot signal thus demonstrating that the in vitro transcripts were indeed introduced into the cell (Supplementary Figure 7b). Subsequent to electroporation the cells were incubated under nutrient deprivation conditions and finally transferred back to full media for recovery. $^{35}$S-methionine incorporation was then assessed during the stress recovery phase. Under these conditions we observed a noticeably increased translational activity in the presence of the tRNA$^{Thr}$ 3′ half of about 35% (Fig. 5b). As previously seen in vitro, addition of CCA or GGU to the 3′ end of the tRNA half eliminated translation stimulation, thus showing the specificity of the tRNA$^{Thr}$ 3′ half effect. While the length of the tRNA$^{Thr}$ 3′ half appears to be critical, the chemical identity of the 5′ end seems to be less crucial. Halves containing a mono- or a triphosphate were equally able to stimulate translation in vivo (Fig. 5c). Similar to the in vitro translation assay, the tRNA$^{Ala}$ 5′ half had no effect in vivo (Fig. 5b).

tRNAs carry multiple post-transcriptional nucleoside modifications whose biological roles in translation and beyond are only beginning to be understood (ref. [18] and references therein). Even the regulatory role of a tRNA-derived fragment in human stem cells has recently been shown to depend on a post-transcriptional modification[19]. The involvement of modified nucleosides in the biology of tRNA-derived fragments however cannot be generalized. The fact that in vitro transcribed tRNA$^{Thr}$ 3′ halves stimulate translation in vivo (Fig. 5b, c) excludes pivotal roles of modifications on that molecule for this particular function. Support for this conclusion comes from LC-MS/MS-based analyses of affinity purified tRNA$^{Thr}$ from unstressed and starved *T. brucei* cells by a neutral-loss scan approach[20]. This RNA mass spectrometry data showed highly similar modification profiles without significant differences (Supplementary Figure 8). Thus these findings do not leave any basis to suspect tRNA$^{Thr}$ nucleoside modifications for being fundamental for the stress-dependent production of the 3′ half and for the observed stimulatory role during translation in vivo and in vitro.

Since both tRNAs and ribosomes are highly conserved components of the translational machinery, we next tested if

stimulation by this 3′ extension is CCA sequence dependent or simply length dependent, we repeated the assay with a tRNA$^{Thr}$ half carrying the sequence GGU at its 3′ end. Also this molecule was unable to stimulate protein synthesis in vitro thus pointing towards a length limitation effect (Fig. 5a). Furthermore, the addition of the tRNA$^{Ala}$ 5′ half or the 3′ half originating from tRNA$^{Asp}$ had no influence on in vitro protein synthesis (Fig. 5a, Supplementary Figure 5). Both of the latter results serve as specificity controls for the stimulatory effect of the tRNA$^{Thr}$ 3′ half.

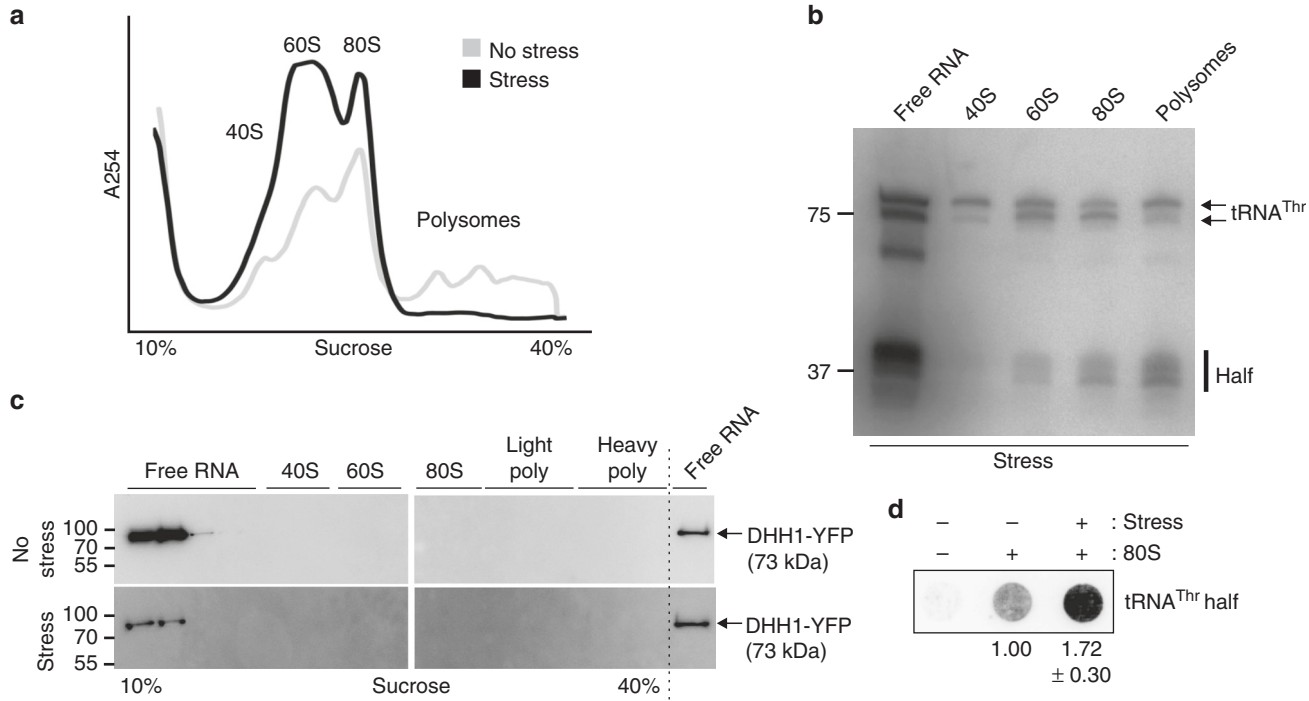

**Fig. 4** The tRNA$^{Thr}$ 3′ half associates with ribosomes in vivo and in vitro. **a** Polysome profiling with sucrose gradients (10–40%) using total cell lysates prepared from exponentially growing *T. brucei* cells (gray) and from cells starved for 2 h in PBS (black). **b** RNA was extracted from fractions containing polysomes, 80S monosomes, 60S, or 40S ribosomal subunits and from the light sucrose gradient fractions (free RNA) and used for northern blot analysis to monitor the tRNA$^{Thr}$ 3′ half (length: 37 nucleotides). In **a** and **b** representative data of three independent experiments are shown. **c** Fractions from the polysome profiles of unstressed (no stress) or starved (stress) cells were also investigated for the presence of the YFP-tagged DHH1 (73 kDa) by western blot analysis. The location of molecular weight protein markers are indicated (see also Supplementary Figure 16). Western blots have been repeated twice. **d** Association of the tRNA$^{Thr}$ 3′ half to ribosomes was analyzed employing in vitro filter binding assay using gradient-purified 80S ribosomes isolated from starvation stressed or unstressed cells and 5′ [$^{32}$P]-end labeled synthetic tRNA halves. The mean and standard deviation of four independent binding experiments are shown underneath the scan, whereas binding efficiency employing unstressed 80S ribosomes was set to 1.00

the tRNA$^{Thr}$ half-mediated translation stimulation can also be observed in other species. Therefore, we introduced the synthetic *T. brucei* tRNA$^{Thr}$ half either into the halophilic archaeon *Haloferax volcanii* or the yeast *S. cerevisiae*. In both organisms we did observe an analogous stimulation in protein synthesis (Supplementary Figure 9a). Also mammalian protein synthesis is stimulated by this tRNA half as exemplified by HeLa extract-based in vitro translation reactions (Supplementary Figure 9b). While we do not have evidence that a similar endogenous tRNA-fragment exists in *H. volcanii*, *S. cerevisiae* or human cells, the mere fact that the *T. brucei* tRNA$^{Thr}$ half functions in distantly related species argues for a highly conserved mode of action.

To gain first mechanistic insight into tRNA$^{Thr}$ 3′ half function, we tested whether this rancRNA stimulates translation by promoting mRNA binding to the ribosome. To this end *T. brucei* in vitro translation reactions were assembled, protein synthesis stopped after 5 or 20 min by the addition of cycloheximide and the ribosomes subsequently pelleted through a sucrose cushion. Northern blot analyses on the ribosome pellet and the corresponding supernatant fractions demonstrated enhanced levels of the highly abundant tubulin mRNA on ribosomes (between 1.3 and 3.8-fold) in the presence of the tRNA$^{Thr}$ 3′ half (Fig. 5d). Concomitantly, tubulin mRNA levels decreased in the ribosome-free supernatants thus hinting at enhanced translation initiation in the presence of the tRNA half.

**Depletion of the endogenous tRNA$^{Thr}$ half eases stimulation.** To study the effect of the endogenous tRNA$^{Thr}$ 3′ half on protein

synthesis we performed affinity purification using a biotinylated antisense DNA oligonucleotide. As input the size-selected small RNA pool (30–40 nucleotides) of *T. brucei* cells was utilized (Fig. 6a). Subsequently the affinity purified material, which is enriched in tRNA$^{Thr}$ 3′ half, and the flow through, which represents the small RNA pool depleted of this tRNA half (Fig. 6b), were tested in in vitro translation reactions. Addition of the affinity purified tRNA$^{Thr}$ half to in vitro translation reactions showed a very mild stimulatory effect (Fig. 6c). The lack of significant translation stimulation under these conditions is due to the considerably low concentration of retrieved endogenous tRNA$^{Thr}$ 3′ half molecules (~1 pmol per reaction as compared to 500 pmol in regular in vitro translation reactions; Supplementary Figure 10). However, when the endogenous pool of small RNAs depleted from the tRNA$^{Thr}$ 3′ half was used in in vitro translation assays we observed a marked inhibition of translation (Fig. 6c). This was solely due to the lack of the tRNA$^{Thr}$ 3′ half as the control fraction (fraction III; - biot-ASO in Fig. 6c) did not show such an effect. These data suggest that in *T. brucei* cells the tRNA$^{Thr}$ half counteracts the inhibitory effects of other small RNAs, such as other tRNA-derived fragments that might be present in this size range, on protein biosynthesis. In vivo support for this interpretation comes from experiments introducing anti-sense oligonucleotides (ASO) targeting the endogenous tRNA$^{Thr}$ 3′ half. Electroporation of these ASO into *T. brucei* neutralizes the role of the endogenous tRNA$^{Thr}$ half resulting in a clear translation inhibition during stress recovery as measured in the metabolic labelling assay (Fig. 6d).

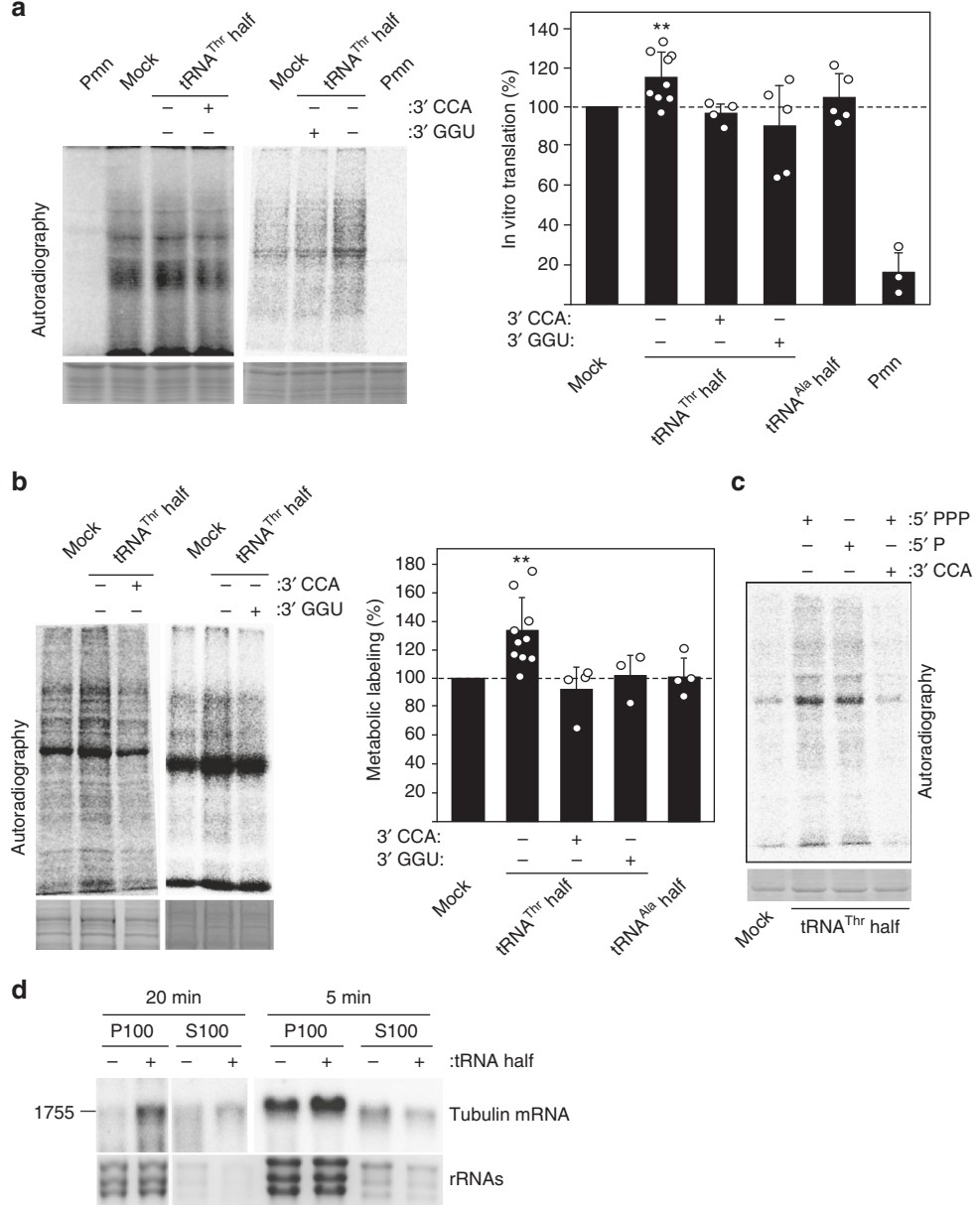

**Fig. 5** The tRNA$^{Thr}$ 3′-half stimulates translation in vitro and in vivo. **a** On the left, the autoradiographs of two representative SDS polyacrylamide gels of in vitro translation assays performed in the absence (mock) or in the presence of in vitro transcribed tRNA$^{Thr}$ 3′ half, either containing a 3′-CCA or 3′-GGU end (+) or lacking it (−), are shown. The mean and standard deviations of four to nine independent in vitro translation experiments in the absence (mock) or presence of tRNA halves (either originating from tRNA$^{Thr}$ of tRNA$^{Ala}$) are shown on the right graph. Addition of the translation inhibitor puromycin (Pmn) serves as specificity control for the assay ($n = 3$). **b** On the left, the autoradiograph of two representative gels of in vivo translation reactions performed in the absence (mock) or in the presence of electroporated tRNA$^{Thr}$ 3′ halves, either containing a 3′-CCA or 3′-GGU end (+) or lacking it (−), are shown. Quantification (mean and standard deviation) of three to ten independent metabolic labeling experiments in the absence (mock) or presence of introduced tRNA halves (either originating from tRNA$^{Thr}$ of tRNA$^{Ala}$) is shown on the right graph. **c** Autoradiograph of an in vivo translation assay using electroporated tRNA$^{Thr}$ 3′ halves containing different chemical groups at the 5′ end. A representative gel of in total four independent experiments is shown. 5′-PPP: 5′ triphosphate; 5′-P: 5′ monophosphate. 3′-CCA: indicates the presence or absence of a 3′ CCA tail. Significance in **a** and **b** according to paired Student's $t$-test: **$P \leq 0.01$. **d** Abundance of tubulin mRNA (1755 nucleotides; see Supplementary Figure 17a) associated with ribosomes (P100) during $T.$ $brucei$ in vitro translation reactions in the presence or absence of the tRNA$^{Thr}$ 3′ half was monitored by northern blot analysis ($n = 2$). S100 indicates the respective post-ribosomal supernatants. Reactions were stopped either after 5 or 20 min of incubation. In all figures either Coomassie stained protein gels or EtBr staining of RNA gels (bottom panels) serve as loading controls

## Discussion

Kinetoplastids, including $T.$ $brucei$, possess a very peculiar RNA biology. For example, most mitochondrial mRNAs are highly edited by insertions/deletions of uridines, all tRNAs need to be imported into mitochondria, certain tRNAs are post-transcriptionally processed by interdependent modification/ editing systems[21], cytoplasmic ribosomes are composed of several distinct large ribosomal subunit rRNA fragments[22], or poly-cistronic primary mRNA transcripts are processed into mature mRNAs via a process called trans-splicing (reviewed in ref. [23]). All these RNA particularities are orchestrated by dedicated processing/editing/transport machineries thus rendering them as

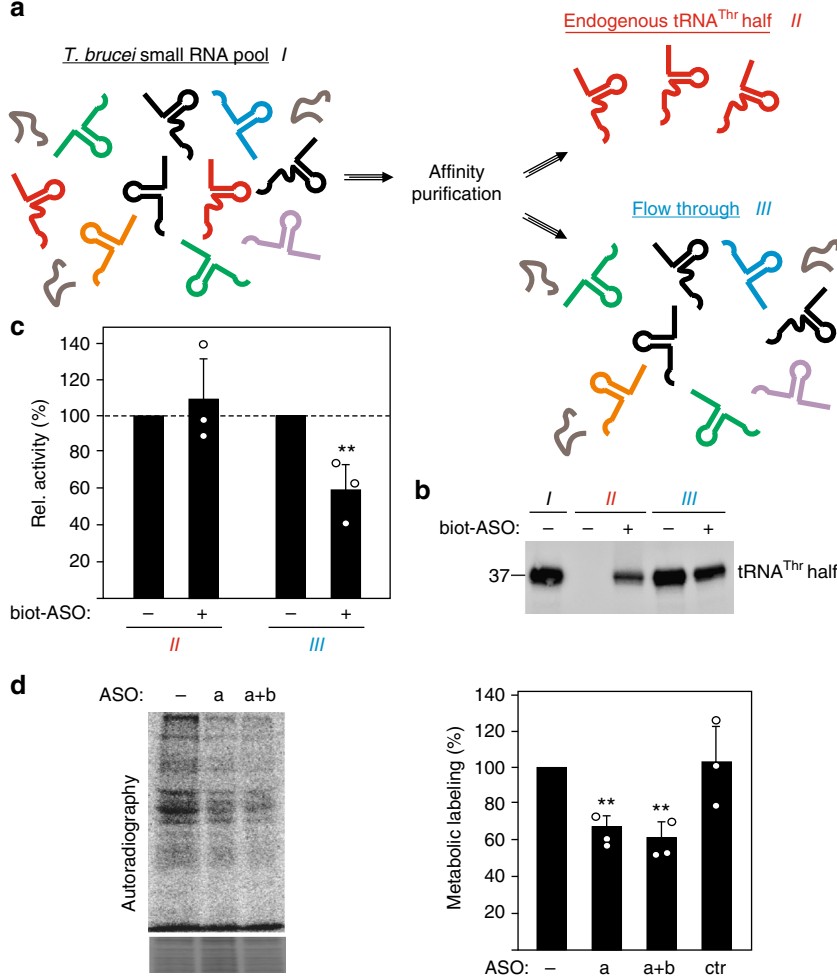

**Fig. 6** Depletion of endogenous tRNA$^{Thr}$ 3′ halves alleviates the stimulatory effects during translation. **a** Scheme of the strategy used for affinity purification of the endogenous tRNA$^{Thr}$ 3′ half from the pool of small RNAs (size range between 30–40 nucleotides). As a control the same procedure was performed in the absence of a biotinylated antisense oligonucleotide (ASO). **b** Northern blot analysis of RNA extracted from the samples obtained after affinity purification showing the successful isolation of the tRNA$^{Thr}$ 3′ half (see also Supplementary Figure 10) when using the biotinylated ASO (fraction II+) and its partial depletion from the corresponding flow through (compare fractions III, − and +ASO). A representative blot of in total two independent affinity purification experiments is shown. **c** Depletion of endogenous tRNA$^{Thr}$ 3′ halves from the pool of small RNAs results in reduced in vitro translation activities. The effects of RNA fractions obtained after affinity purification were tested on in vitro translation. Quantification (mean and standard deviation) shows the average of three independent experiments. **d** Chemically modified ASO complementary to the tRNA$^{Thr}$ 3′ half were electroporated into *T. brucei* cells and their effect on in vivo translation was investigated by metabolic labeling during stress recovery. The autoradiograph shows a representative SDS polyacrylamide gel in which metabolic labeling was performed in cells electroporated without an ASO (−) or cells electroporated with one (a) or two (a+b) different ASO targeting endogenous tRNA$^{Thr}$ 3′ halves. Coomassie staining of the gel (bottom panel) serve as loading control. Quantification of three independent metabolic labeling experiments is shown on the right. As specificity control metabolic labeling was performed also after electroporation of an analogous ASO without any sequence complementarity to the tRNA$^{Thr}$ 3′ half (ctr). Significance according to paired Student's *t*-test: **$P \leq 0.01$

potential targets for therapeutic interventions. We have recently identified the class of ribosome-associated small ncRNAs (rancRNAs) as a thus far unknown family of translation regulators[9]. RancRNAs directly associate with the translation machinery and due to their small size and immediate availability possess the potential of being first-wave regulators during stress encounters[8]. To gain insight into *T. brucei* rancRNA biology we sequenced and analyzed a rancRNA library originating from cells grown to exponential or stationary phases (procyclic as well as bloodstream form) and from temperature-stressed or starved cells. It became evident that especially tRNA-derived reads increase significantly upon starvation and that most of them represent tRNA halves (Fig. 1a, b).

One of the most abundant tRNA-derived species originates from the 3′ half of tRNA$^{Thr}$ during nutritional stress and during

the stationary phase in the procyclic form (Fig. 1d, Supplementary Figure 1, Supplementary Data 1). The production of tRNA halves by cleavage in or around the anticodon loop during various stress conditions has been observed before in different model systems including Trypanosomatids[24], ciliates[25], and mammalian cells[26–29]. In the kinetoplast *T. cruzi* tRNA halves were also detected during unstressed conditions[30] but became more abundant under nutritional stress[24]. While uncovering the physiological role of these tRNA fragments awaits further dedicated research, it could be shown that in *T. cruzi* these stress-induced tRNA halves accumulate in cytoplasmic granules. In subsequent studies, it was shown that a certain amount of these tRNA halves are secreted to the growth medium in extracellular vesicles[31,32] thereby potentially delivering regulatory ncRNA molecules to other parasites as well as to mammalian cells. A similar role for

exosome-packed small RNA molecules, including tRNA halves, has been recently suggested in protozoan parasites of the genus *Leishmania*[33]. Analogous insight into the tdR RNome and its putative function in the closely related parasite *T. brucei* is largely missing[34]. In variance to the tRNA halves pool in *T. cruzi*, which seems to be dominated by halves originating from only a few tRNA isoacceptors[24,30], the *T. brucei* tRNA halves population is much more diverse (Supplementary Figure 1, Supplementary Data 1).

While in most reported cases the biological role and the cellular target of the sequenced tRNA half molecules remained elusive, some studies clearly demonstrated a function relevant for stress response. In human cells it was shown that the stress-induced tRNA$^{Ala}$ 5′ half inhibited protein synthesis by sequestering certain translation initiation factors and promotes stress granule formation[13,28]. The tRNA$^{Thr}$ 3′ half identified here functions in a markedly different manner. It is produced during starvation conditions or when procyclic *T. brucei* cells enter stationary phase and it binds to ribosomes and polysomes (Fig. 4). However, instead of inhibiting the translation machinery, as it has been shown for other rancRNAs[6–8], this ribosome-bound RNA stimulates protein biosynthesis (Fig. 5). Particularly after prolonged starvation, which leads to a complete loss of cell motility, the tRNA$^{Thr}$ 3′ half is produced, associates with ribosomes and stimulates protein synthesis during the stress recovery phase. Blocking of endogenous tRNA$^{Thr}$ halves via an antisense oligonucleotide (ASO) approach or depleting the tRNA$^{Thr}$ halves by affinity purification eases this stimulatory effect on protein biosynthesis in vivo and in vitro (Fig. 6). While the tRNA$^{Thr}$ half is special in the sense that it is the only 3′ half identified to robustly associate with ribosomes (Fig. 1d), structurally it does not show any obvious peculiarities compared to other identified tRNA-derived fragments. The aforementioned human tRNA$^{Ala}$ 5′ half, demonstrated to inhibit translation initiation and promote stress granule formation, was shown to form G-quadruplexes as prerequisite for its biological function[35]. The *T. brucei* tRNA$^{Thr}$ 3′ half however does not seem to adopt G-quadruplex structures (Supplementary Figure 11) thus hinting at a different mode of action.

Interestingly in the bloodstream form of *T. brucei*, the predominant form of the parasite within the mammalian host, this tRNA half is present in easily detectable amounts throughout all growth phases or stress situations (Fig. 2). This suggests that during this life stage the metabolism of *T. brucei* can more strongly benefit from the presence of this rancRNA. It is known that the doubling time of *T. brucei* bloodstream forms is significantly shorter compared to the insect form[36], which could be in part related to the stimulatory effect on metabolism of the tRNA$^{Thr}$ 3′ half. In this context it would be of great interest to investigate the situation in other life stages, namely in the non-dividing forms stumpy and metacyclic trypanosomes.

How can a rancRNA stimulate the ribosome during protein production from a mechanistic point of view? When investigating a specific endogenous mRNA we demonstrated enhanced association of the tubulin mRNA with ribosomes in the presence of the tRNA$^{Thr}$ 3′ half (Fig. 5d). These findings are compatible with the view that the tRNA half stimulates translation by promoting the initiation phase in *T. brucei*. In order for the tRNA$^{Thr}$ 3′ half to fulfill this role in vitro and in vivo, the 3′ CCA end needs to be removed (Fig. 5a, b). We have shown that almost all tRNAs rapidly lose their CCA end upon starvation (Supplementary Figure 1a). In the presence of 3′ extensions on the half, CCA or the artificially added GGU, the translation stimulatory effect is lost. This opens the possibility for a straightforward regulation of the tRNA$^{Thr}$ 3′ half activity in vivo by modulating the extent of 3′ CCA tail addition/removal. Comparing northern blot signals for

the tRNA$^{Thr}$ 3′ half to synthetic standards revealed about 2500 molecules per stressed cell. Therefore the tRNA$^{Thr}$ 3′ half is approximately two orders of magnitude less abundant than the expected ribosome pool in *T. brucei*. However, one can envision a role of this tRNA half acting in a turnover mode. Our affinity purification experiments demonstrated that depletion of the tRNA$^{Thr}$ half from the pool of small RNAs in the size range between 30–40 nucleotides actually inhibits global translation (Fig. 6c). It is therefore conceivable that the tRNA$^{Thr}$ 3′ half counteracts any inhibitory small RNA in this size range (including other potential tRNA-derived RNAs) present in a stressed cell. We have shown before in *S. cerevisiae* and *H. volcanii* that rancRNAs can have global inhibitory effects on protein production in vivo despite the fact that they are less abundant than the ribosome pool[6,8]. Thus even though we do not have uncovered the complete molecular mechanism used by the tRNA$^{Thr}$ 3′ half to boost translation in *T. brucei* during stress recovery, our data are compatible with a stimulatory role during mRNA loading (Fig. 5d). In this scenario, the tRNA$^{Thr}$ half-occupied ribosomes (a complex which accumulates during nutrient deprivation (Figs. 2a–c and 4b)) are more efficient recommencing translation, once the parasite encounters more favorable environmental conditions. Based on the presented SDS gels (Fig. 5a–c), the tRNA$^{Thr}$ half appears to affect global translation rather than specific mRNAs.

In human cells stress-induced tRNA halves are known to be produced by angiogenin, an RNase A-type enzyme that cleaves tRNAs in the anticodon loop[25,28]. *T. brucei* does not possess an angiogenin homolog, implying an alternative tRNA half biogenesis pathway in the parasite. In humans and plants it has been reported that tRNA-derived RNA fragments (tdRs) can also be produced by Dicer, a central endonuclease of the si/miRNA machinery[37,38]. Since *T. brucei* does possess the siRNA pathway albeit lacking miRNAs, we investigated whether the Dicer homologs might be involved in processing of the tRNA$^{Thr}$ 3′ half. Utilizing RNAi against the two dicer-like proteins (TbDCL1 and TbDCL2) we could exclude these nucleases as being involved in tRNA cleavage in *T. brucei* (Supplementary Figure 12). In yeast Rny1, a member of the RNase T2 family, has been shown to cleave tRNAs in the anticodon loop during oxidative stress thus producing half molecules[39]. However, no Rny1 homolog has been identified in the genomes of *T. brucei* or its close relatives. It is remarkable that the involvement of tdRs in regulating cellular processes is evolutionarily so conserved (it can be observed in all domains of life), yet their biogenesis seems to involve a big variety of diverse enzymes. It therefore appears that in Kinetoplastids, a completely different tRNA processing machinery is at work which awaits to be uncovered.

The tRNA$^{Thr}$ 3′ half described herein is the latest addition of functional tdRs reported in recent years. The class of tdRs, which includes tRNA halves as well as also shorter fragments of ~14–26 nucleotides, was initially described in cells and organisms exposed to challenging growth conditions in all three domains of life (reviewed in refs. [40,41]). Subsequent studies revealed tdRs also during normal conditions suggesting possible house-keeping functions (ref. [42] and references therein). Certain tdRs have been recognized as pivotal regulators of cell metabolism, particularly during cellular stress and disease (reviewed in refs. [41,43]). Unlike other small ncRNA regulators (such as miRNA, siRNAs or piRNAs) tdRs are a structurally and functionally highly multifaceted class of ncRNAs[40]. The so far identified biological roles of tdRs include regulation of transcription, translation, stress granule formation, apoptosis, cell proliferation, RNAi, vesicle-mediated intercellular communication, intergenerational inheritance, and retrotransposition (reviewed in ref. [40]). Very recently a 22 nucleotide long tdR has been identified to regulate ribosome

biogenesis in human cells by controlling translation of crucial ribosomal proteins[44]. RNAi against this tdR resulted in impaired cell viability and increased apoptosis in human cancer models. In most of the reported cases on tdR function the tRNA fragment seems to inhibit a cellular process. The *T. brucei* tRNA^Thr 3′ half reported here is one of the few examples in which a tdR stimulates a cellular function, such as translation during stress recovery. This further diversifies the regulatory potential of tdRs as compared to other small ncRNA regulators. Therefore we cannot exclude the possibility that the *T. brucei* tRNA^Thr 3′ half has additional biological roles in the parasite beyond translation control. It is astounding that the "precursor" molecule of tdRs, genuine tRNA that is, has basically one major cellular role as substrate for the protein synthesis machinery, while processing products thereof are functionally so heterogeneous. Thus post-transcriptional cleavage events can generate novel regulatory molecules thereby further increasing the complexity of cellular RNomes in general and expanding tRNA biology in particular.

## Methods

**Strains and growth conditions**. *Trypanosoma brucei* procyclic stage (PCF) 427, 29–13 or bloodstream forms New York single markers (NYSM)[45] cell lines were used in all experiments. Procyclic stage cells 427 and 29–13 were grown at 27 °C in SDM-79 media supplemented with 5 or 10% fetal calf serum (FCS), respectively[46]. Bloodstream forms (BSF) were cultured in HMI-9 medium containing 10% FCS at 37 °C/5% CO₂ (ref. [47]). Cultures were harvested in the exponential growth phase at densities lower than $2 \times 10^7$ cells/ml for PCF and $10^6$ cells/ml for BSF. For stationary phase procyclic cells were harvested at a density of $6–7 \times 10^7$ cells/ml. For heat shock, exponentially growing cells were incubated 30 min in 41 °C pre-warmed medium; cold shock was applied to PCF cells by incubating them for 30 min in 13 °C media; oxidative stress: media containing 125 μM of oxygen peroxide (H₂O₂) for 1 h. Unless stated otherwise, nutritional stress was applied to PCF cells by incubating them for 2 h in 1x PBS (phosphate buffered saline) at 27 °C. Nutritional stress of BSF involved incubation for 1 h in 1x PBS at 37 °C. A cell line expressing an endogenous copy of DHH1 N-terminally tagged (eYFP) was constructed using a plasmid kindly provided by Kramer and Carrington[15]. Transfection, cloning and selection of transgenic 427 procyclic cells were done as described[48].

**RancRNA library preparation and bioinformatics analyses**. Ribosome-associated small RNAs were isolated as described[49]. The cDNA library was prepared using the TruSeq Small RNA Library Prep kit (*Illumina*) and sequenced using the Illumina HiSeq 2000 platform. Bioinformatics analysis was performed as described by Luidalepp et al.[50] with the exception that four mismatches were allowed. Overlapping read pairs were joined using Pandaseq[51]. Next, reads were mapped to the *T. brucei* TREU927 reference genome obtained from TriTrypDB[52] using STAR[53]. The genome annotation in version 32 of TriTrypDB was complemented with tRNA gene prediction using tRNAscan-SE[54] and transcript features were extended by 50 nucleotides up and downstream to include tRNA precursor regions. Subsequently, reads mapping within annotated transcripts were extracted and putative processing products were identified with a modified version of the APART pipeline[10]. In particular, for the identification of RNA processing events, read blocks as defined using the blockbuster algorithm[55] were used instead of contigs composed of overlapping reads. The normalization of expression levels across samples was performed using the Bioconductor and edgeR package[56]. The statistical analysis was performed in the R environment. All sequencing reads were submitted to the European Nucleotide Archive (ENA) and can be accessed with the number PRJEB24915.

**Electroporation of procyclic *T. brucei* cells**. Exponential growing 427 cells ($4.5 \times 10^7$) were harvested by centrifugation ($1400 \times g$ for 10 min at 4 °C) and resuspended in 1 ml of ice cold 1x Cytomix (25 mM HEPES/KOH pH 7.6, 10 mM K₂HPO₄, 120 mM KCl, 0.15 mM CaCl₂, 5 mM MgCl₂, 2 mM EDTA). Cells were washed with 1 ml 1x Cytomix, resuspended in 190 μl 1x Cytomix and mixed with 500 pmol of in vitro transcribed tRNA half in 1x annealing buffer (10 mM Tris-HCl; pH 7.6, 80 mM MgCl₂) to a final volume of 200 μl. The mixture was electroporated twice using a Bio-Rad gene pulser II (1.2 kV, 25 μF, and 0 Ohm) in a 4 mm electroporation cuvette (EP-104, *Cell Projects Ltd.*). Finally, the procyclic *T. brucei* cells were resuspended with 1 ml of SDM-79 (5% FCS) and transferred into pre-warmed 3 ml medium for a 2 h recovery at 27 °C. The following RNA strands were produced by in vitro transcription using T7 RNA polymerase[57,58] and subsequently introduced with this approach into *T. brucei*: Thr_(AGU)-3′-half + CCA: 5′ AAGACGGAGGUCGGGGGGUUCGAUCCCCCAGUGGCCUCCA 3′, Thr_(AGU)-3′-half-CCA: 5′ AAGACGGAGGUCGGGGGGUUCGAUCCCCCA GUGGCCU 3′, Thr_(AGU)-3′half + GGU: 5′ AAGACGGAGGUCGGGGGUUC

GAUCCCCCAGUGGCCUGGU 3′, Thr_(AGU)-5′-half: 5′ GGCCGCUUAGCU CAAUGGCAGAGCGCCGUCCUAGU 3′. The tRNA^Ala 5′ half used was a synthetic oligonucleotide (*Microsynth*) with the sequence Ala_(CGA)-5′-half: 5′ GGGGAUGUAGCUCAGAUGGUAGAGCGCCCGCUUAGC 3′.

**Metabolic labeling**. Translation activity in *T. brucei* was monitored by metabolic labeling. $4.5 \times 10^7$ procyclic cells were electroporated with the tRNA halves (500 pmol) as described above, allowed to recover for 2 h under normal growth conditions and finally stressed by incubation in 1x PBS. After 2 h of nutritional stress, the cells were again harvested (see above) and resuspended in 750 μl pre-warmed media. 1/3 or the cells were used for RNA extraction and subsequent detection of the electroporated tRNA halves by northern blot analysis. The remaining 2/3 of the cells were mixed with 250 μl SDM-79 (5% FCS) containing 2 μl of L-³⁵S-methionine (10 μCi/μ, *Hartmann Analytic*) and incubated for 60 min at 27 °C. After metabolic labeling the cells were harvested, resuspended in 1x Laemmli buffer and proteins were separated by 10% SDS-PAGE. Radiolabeled methionine incorporation was measured by phosphorimaging. Metabolic labeling in *H. volcanii* and *S. cerevisiae* was performed as described previously[6,8].

**Inactivation of tRNA^Thr 3′ half by ASOs**. To block the endogenous tRNA^Thr 3′ half, modified ASOs were used (ASO_a: mG*mA*-mA*mC*mC*C*C*C*C*G*A*C*T*C*C*mG*mT*mC*mT*mT and ASO_b mA*mG*mG*mC*mC*A*C*T*G*G*G*G*G*A*mT*mC*mG*mA*mA, *Qiagen*). As specificity control an analogous ASO strand with a completely unrelated sequence was used (ASO_ctr: mG*mU*mA*mU*mU*T*A*-C*A*A*T*T*G*A*C*mG*mU*mA*mU*mA). All ASOs were designed as RNA/DNA/RNA chimeras with a phosphorothioate backbone (asterisks). The ten central deoxyribonucleotides are flanked by five 2′-O-methyl modified ribonucleotides. 500 pmol of the ASO were electroporated into $4.5 \times 10^7$ cells. After two hours of recovery in normal media nutritional stress was applied for four hours and global translation was assessed during a two-hour recovery period by metabolic labeling as described.

**Cell extract preparation**. Cells grown under different conditions were harvested (see above) and the cell pellets ($1–3 \times 10^9$ cells) resuspended in 500 μl ribosome buffer A (120 mM KCl, 20 mM Tris/HCl pH 7.6, 2 mM MgCl₂, 1 mM DTT) containing 20 mM ribonucleoside vanadyl complex (RVC, *Bioconcept*) and 2.5 μl of RiboLock (40 U/μl, *Thermo Scientific*) and flash frozen. Samples were passed 10 times through a 25G needle and 10 times through a 27G needle and extracts were cleared by centrifugation. Cell extracts were aliquoted, snap frozen and stored at −80 °C until use.

**T. brucei in vitro translation**. One in vitro translation reaction with a total volume of 30 μl is composed of 6 μl "translation mix" and 24 μl "sample mix". The 6 μl translation mix contained 3 μl 10 x translation cocktail (100 mM Hepes/KOH pH 7.4, 15 mM Mg(OAc)₂, 750 mM KOAc, 4 mM GTP, 10 mM ATP, 500 μM of each amino acid except methionine), 0.5 μl of Mg(OAc)₂ (100 mM), 0.5 μl of creatine phosphokinase (10 mg/ml, *Roche*), 1 μl of creatine phosphate (0.6 M) and 1 μl of L-³⁵S-methionine (10 μCi/μl). The 24 μl sample mix was composed of 3 μl DMSO, 7 μl *T. brucei* cell extract (extract prepared as described above and this volume corresponds to $2–4 \times 10^7$ cell equivalents) containing 20 mM RVC, 7.5 μl ribosome buffer B (120 mM KCl, 20 mM Tris/HCl pH 7.6, 8 mM MgCl₂), 500 pmol tRNA halves in annealing buffer (10 mM Tris/HCl pH 7.6 and 20 mM NaCl). The reaction was started by combining the translation mix with the sample mix and proceeded for 20 min at 27 °C. The reaction was stopped by addition of 1x Laemmli buffer and incubation at 95 °C for 2 min. Proteins were subsequently separated on a 10% SDS-PAGE gel and methionine incorporation was monitored by phosphorimaging. For testing the tRNA^Asp 3′ half (anticodon GUC) during in vitro translation, the in vitro transcribed RNA strand 5′ CACGCGGGUGACCCGGGU UCAAUUCCCGGCCGGGAAGCCA 3′ was used as above. To test the effect of endogenous tRNA^Thr 3′ half 5 μl (~1 pmol) of affinity purified samples (or the equivalent volume of the corresponding flow through) were used with 3.5 μl cell extract prepared with RVC, 2.5 μl ribosome buffer B in a total volume of 10 μl. Subsequently 3 μl of translation mix was added. The reaction was incubated and stopped as described above.

To assess the mRNA association to *T. brucei* ribosome, the equivalent to four in vitro translation reactions (using cold L-methionine) were run as described above and stopped after 5 or 20 min by the addition of cycloheximide (final concentration 100 μg/ml). Samples were loaded onto 800 μl of a 1.1 M sucrose cushion prepared in ribosome buffer A containing 1 mM DTT and 100 μg/ml cycloheximide. After centrifugation (2.5 h at $200,000 \times g$, 4 °C) the supernatant (S100) was recovered, 2.5 vol. 100% ethanol was added and incubated overnight at −20 °C. Samples were centrifuged ($16,000 \times g$, 45 min, 4 °C), the pellets resuspended in 1 ml of trizol and RNA was extracted following the manufacturer's instructions. RNA was also extracted from the pellet fraction (P100) of the $200,000 \times g$ centrifugation step (see above) using trizol and used for mRNA northern blotting (see below).

**Affinity purification of tRNA^Thr 3′ half**. To affinity purify the endogenous tRNA^Thr half the cell lysates obtained from nutritionally stressed cells were first incubated for 1 h at room temperature prior to total RNA isolation. Hybridization to a complementary DNA strand was done by incubating 1 μl (100 pmol) of a 3′ end-biotinylated antisense DNA oligo (biot-ASO) with 5 μg of size-selected (30–40 nt) total RNA in 100 μl of 5x SSC buffer (750 mM NaCl, 75 mM trisodium citrate). The sample was denatured for 3 min at 90 °C followed by hybridization for 10 min at 65 °C. The RNA-DNA hybrid was immobilized onto pre-washed Streptavidin magnetics beads (25 μl, *Roche diagnostics*) and incubated for 30 min at room temperature with rotation. The supernatant containing the unbound RNA pool (flow through) was removed and stored for future use. Beads were washed once with 50 μl 1 x SSC buffer and 3 times with 50 μl of 0.1 × SSC buffer. The tRNA^Thr half was eluted by heating the beads in 100 μl of water at 75 °C for 3 min. Contaminating biot-ASO were removed by DNase I treatment (*Thermo Scientific*). The affinity purified tRNA half was extracted by using Roti®-Aqua-P/C/I (*Roth*). Finally, the RNA of the flow through and the affinity purified tRNA half were precipitated with 2.5 volumes of 100% EtOH at −20 °C overnight. The samples were washed with 70% EtOH and resuspended in 15 μl water. To verify the quality of the affinity purification procedure 200 ng of RNA was loaded onto a denaturing 8% polyacrylamide gel followed by northern blot analysis. The sequence of the complementary DNA oligonucleotide (biot-ASO) used for affinity purification was 5′ AAGCCACTGGGGGGATCGAACCCCCGACCTCCGTCTTACTAGGACG GCGCTCTGCCATTGAGCTAAGCGGCCAAA 3′ (*Microsynth*). The 3′ AAA overhang was added for better binding to the beads. As control, the entire affinity purification procedure was performed in the absence of biot-ASO as well.

**Polysome profiling**. For polysome profiling, cell extracts were prepared as previously described without the addition of RVC from exponentially growing or nutritionally stressed cells. 50–100 OD_260 of cell extract was layered on top of a 10–40% (w/v) sucrose gradient prepared in ribosome buffer A in SW 32Ti tubes (Beckman Polyallomer Centrifuge tubes 25 × 89 mm). The gradients were centrifuged in a Beckman SW 32Ti rotor (6 h at 25,000 rpm at 4 °C). Gradients were pumped out and fractions were collected every 16 s while continuously monitoring the absorbance at 260 nm. For downstream northern blot analyses the desired fractions were pooled and precipitated with EtOH before RNA extraction with 1 ml of TRI Reagent (*Zymo Research*). For 80S ribosome preparation, the fractions containing monosomes were pooled into a Beckman Optiseal Polyallomer tubes (volume 32.4 ml) and filled up with 1x ribosome buffer A. The ribosomes were pelleted by ultracentrifugation at 33,000 rpm (100,000 × $g$) for 17 h at 4 °C (rotor type 60 Ti, *Beckman*). After centrifugation the pellet was resuspended in 200 μl ribosome buffer A. The concentration of ribosomes was determined by the absorption at 260 nm (1 A_260 = 18 pmol 80S).

**Northern blot analyses**. For northern blotting, 3–40 μg total RNA extracted with TRI Reagent (*Zymo Research*) according to the manufacturer's protocol was complemented with 1 volume of 2x RNA loading dye and separated on an 8% denaturing polyacrylamide gel (7M Urea, in 1x TBE buffer). The gel was subsequently electroblotted onto a nylon membrane (Amersham Hybond N⁺, *GE Healthcare*) as described[7]. See Supplementary Methods for a full list of DNA oligonucleotides used.

For tubulin mRNA northern blot analyses 1.6–3.1 μg RNA of the S100 and 4.4–6.0 μg RNA P100 fractions (see in vitro translation procedure above) were run on a 1% agarose gel in 20 mM MOPS buffer (pH 7.0) containing 1.5% formaldehyde. Subsequently the RNA was blotted onto a nylon membrane (Amersham Hybond N⁺, *GE Healthcare*) by passive transfer. DNA probes were prepared from gel-purified PCR products of the tubulin gene and radioactively labelled by using the Prime-a-Gene labelling protocol (*Promega*). 2.5 μl of a mixture of random hexamers (100 μM; *Thermo*) and 5 ng of PCR product in a total volume of 17.5 μl were heated to 95 °C for five minutes to dissociate the dsDNA. After reanneal the random hexamer primers were extended by 1.5 μl of Klenow polymerase (3 U) by the addition of 2.5 μl Klenow buffer, 1 μl of dNTPs (20 μM final concentration dATP, dTTP, dGTP), 2.5 μl of [α-³²P]CTP for 1 h at 37 °C. The reaction was stopped by adding 2 μl of 0.5 M EDTA and 70 μl H_2O. The northern blot probe was heated to 95 °C for five minutes and subsequently added to the prehybridized membrane. See Supplementary Figures 13–17 for uncropped blots and full-length gels.

**Western blot analyses**. Fractions (16 s) were collected from a polysome profiling experiment of either exponential or PBS stressed DHH1-YFP tagged cell lysate (60 OD_260 cell extract per gradient). Ten microlitres of every third fraction was mixed with 2x Laemmli buffer, denatured at 95 °C for 2 min, loaded on a 10% SDS-PAGE gel, and run for 1 h at 160 V. The gel was transferred onto a nitrocellulose membrane (*Amersham Biosciences*) and blocked for 1 h in 1x PBS containing 0.1% Tween-20 in 5% nonfat dry milk. The membranes were incubated with mouse anti–GFP antibody (1:1000, *Roche*; cat. number: 11814460001) at 4 °C overnight. After washing (3 × 10 min each with 1 x PBS, 0.1 % Tween-20) horseradish peroxidase conjugated secondary antibodies were added for 1 h at room temperature (1:3000; *Roche*). The membranes were washed as before and results were visualized

using an enhanced chemiluminescence SuperSignal West Femto Maximum Sensitivity Substrate (*ThermoFisher Scientific*).

**In vitro binding studies**. Binding studies of tRNA^Thr 3′ half to ribosomal particles purified from stressed or unstressed cells (as described above) were performed using a dot blot-filtering approach. For the filter binding assay 5 pmol of *T. brucei* 80S ribosomes were incubated with 1 μl of 5′-[³²P]-end-labeled tRNA^Thr half (4 pmol/μl) in 25 μl ribosome buffer B for 30 min at 27 °C. After incubation the reactions were filtered through a nitrocellulose membrane (0.45 μM diameter) using a vacuum device, followed by 2 washing steps with ice cold buffer B. Membranes were exposed to phosphorimaging screens for 1 h and quantified with a phosphoimager.

**Reporting summary**. Further information on experimental design is available in the Nature Research Reporting Summary linked to this article.

## Data availability

All sequencing data generated in this study have been deposited at the European Nucleotide Archive (ENA) and can be accessed with the number PRJEB24915. All other data are available from the corresponding authors on request. A reporting summary for this article is available as Supplementary Information file.

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

## Acknowledgements

We thank Isabel Roditi for the siRNA constructs targeting TbDCL1 and TbDCL2, Mark Carrington and Susanne Kramer for DHH1 constructs, Anna Stocker for experimental help during her bachelor project, Yulia Gonskikh and Bernd Schimanski for help with HeLa in vitro translation and mRNA northerns, respectively. Our thanks are extended to Julia Reuther for yeast expertise. The work was primarily supported by the NCCR 'RNA & Disease' funded by the Swiss National Science Foundation. Additional support from the Swiss National Science Foundation grant 31003A_166527 [to N.P.] is acknowledged.

## Author contributions

R.F. and R.B. conducted the majority of the experiments and analyzed the data. M.F. and L.W. contributed to metabolic labeling experiments. M.Z. and H.L. performed the bioinformatics analyses. R.B., M.C. and O.J. performed all experiments for the revised version of this work. M.H. performed and analyzed the LC-MS/MS experiments. A.S. provided *T. brucei* expertise, analyzed the data and commented on the manuscript. N.P. conceived the study and together with M.C. designed the experiments, supervised the study, analyzed the data and wrote the final version of the manuscript.
