## [Peer Review File · Nature Communications]

Reviewers' Comments:

Reviewer #1:

Remarks to the Author:

This is an excellent report from Fricker et al that deserves publication in Nature Communications upon minor revisions.

This study has several novel aspects that have not been reported in the area of tRNA-derived fragments.

1) Identification of ncRNA-ribosome interactome in *T. brucei* under physiological conditions (life stage and stresses)

2) Unique transnational response to the manipulations with tRNA^{Thr} half: While many different molecular mechanisms of tRNA-derived RNAs are described, they uniformly showed inhibition of protein synthesis. This is the first report where tRNA halves STIMULATE translation in vitro and, what is the most important, in vivo. This is important message

3) Correlation of tRNA halves production with specific physiological condition.

Few points to address:

1. Please include any other 3'tRNA half in your functional analysis. Is stimulation of translation unique to 3'-tRNA Thr half or it can be observed with other 3'halves? Any in vitro assay would be adequate and acceptable

2. Will 3'tRNA thr half stimulate in vitro translation in mammalian system, e.g. in RRL?

3. Supplementary figure 1A (tRNA read length) should be a part of Fig 1 in the main text

4. Authors should include predicted secondary structure of 3'tRNA Thr half into Supplementary information. Is there any particular feature of this tRNA half that stands out (e.g. G-richness, aberrant movement in gels, prediction of G-quadruplex formation)?

5. In response to starvation stress, mature tRNA^{Thr} lacks CCA end. Do other tRNAs or their 3'halves also tend to lack CCA ends under such stress? Discuss in the Results

6.Optional: Addition of CCA-end to tRNA-Thr half limits their ability to stimulate translation. Does this suggest that effect is length-dependent or CCA-dependent? How will a variant of tRNA^{Thr} half with any other triplet behave in in vitro translation assay?

7. Reference 12 is incomplete, should be : "Heat shock causes a decrease in polysomes and appearance of stress granules in trypanosomes independently of eIF2 α phosphorylation at threonine 169, the residue equivalent to serine 51 in mammalian eIF2 α "

Reviewer #2:

Remarks to the Author:

he manuscript by Fricker et al., describes a that tRNA half modulates translation as a stress response

The idea is interesting but the data presented throughout this study is not at the level of publication and especially not in the well respected Nature communication journal. Most of the data was not backed up by statistics. It is not sufficient to present a single RNA-seq experiment and also not to provide the original data that led to the conclusions (see my comments below). However, the main reason that I think this paper does not suit the journal is mainly because there is no model behind how this half tRNA works to affect translation and why CCA affects the phenomenon.

The data in Figure 1

The sequencing of tRNA to assess their level under different conditions is not a trivial task. This is due to the presence of many modifications that makes the RT fall on the way...this is one of the reason why it is easy to identify 3' half of RNA. To overcome this problem and to be able to evaluate changes in the elvel of tRNA is to use the thermostable group II intron reverse transcriptases (TGIRTs) (see Zheng et al. Nature Methods 2015). Because of the modification on tRNA it is not clear if changes in the very distinct population (30 nt) is due to changes in the level of the RNA or due to changes in modification. The data presented in this Figure seems to be done only once and the actual data (RNA-seq) is not presented (In Supp). These days one need to at least repeat such an experiment three times and provide statistical significance of the data.

There is a very rich literature about half tRNA in trypanosomatids that is not mentioned in the paper and these studies also provide information regarding change in half tRNAs. This study does not compare their data to others. In many cases, the half tRNAs are also secreted by the cells via different type of vesicles suggesting that these might be transmitting signals to the parasite population or to the host...(for instance Lambertz et al., 2015)

Indeed half tRNA may function to control innate immunity that is very relevant for infectious organisms.

Figure 2: The explanation why the phenomenon of half tRNA formation during starvation and the inhibition during recovery in the two life stages is not convincing. The difference might be coming from the biological relevance for starvation in the two life stages.

Figure 4: The association of half tRNA should be checked by in situ with the tRNA and DHH1 (IFA or fluorescence). There are studies in trypanosomes showing that half tRNA are found in stress granules (Gracia-Silva Mol. Biochem. Parasitol 2010).

Figure 5: tRNAs are heavily modified it is strange that in vitro transcribed RNA has a biological role...

the CCA at the 3' end may also protect the half tRNA from degradation...

Figure 6: The fact that in vivo selected half-tRNA has only a very minor stimulatory effect is a bit confusing and bothering. Is it possible to quantitate the amount of RNA used in transfection? I think that the main difference between in vivo and in vitro half tRNA is not the amount but the fact that the tRNA are either modified or not...

In sum, the phenomenon is interesting, but as I outlined above a few of the experiments need to be repeated but I think that to publish the study in Nature communication one needs to show the mechanism of the activation

Point-by-point response:

Comments to referee #1:

1. Please include any other 3'tRNA half in your functional analysis. Is stimulation of translation unique to 3'-tRNA Thr half or it can be observed with other 3'halves? Any *in vitro* assay would be adequate and acceptable

Our comments:

Following this referee's suggestion we have now used another 3'-tRNA half in an *in vitro* translation assay. We used the tRNA^{Asp} 3'-half because it is one of the few observed 3'-fragments in our cDNA library and it has a similar expression pattern as the tRNA^{Thr} 3' half (see the **new Supplementary Fig 1b**). As shown in the **new Supplementary Fig. 5** the tRNA^{Asp} 3'-half does not affect *in vitro* translation.

2. Will 3'tRNA thr half stimulate *in vitro* translation in mammalian system, e.g. in RRL?

Our comments:

We have tested the effects of the *T. brucei* tRNA^{Thr} 3' half in a HeLa cell extract-based *in vitro* translation assay. As can be seen in the **new Supplementary Fig. 8b** we do also see reproducible translation stimulation in this case. These additional findings support our conclusion that the mode of action and/or the ribosome binding site of the tRNA^{Thr} 3' half is highly conserved.

3. Supplementary figure 1A (tRNA read length) should be a part of Fig 1 in the main text

Our comments:

Even though we toned down the bioinformatics analyses (based on the critical comments from referee #2; see below) we have followed this reviewer's suggestion and have moved the figure about the tRNA read length distribution into Fig. 1b of the main text.

4. Authors should include predicted secondary structure of 3'tRNA Thr half into Supplementary information. Is there any particular feature of this tRNA half that stands out (e.g. G-richness, aberrant movement in gels, prediction of G-quadruplex formation)?

Our comments:

Following this good suggestion we have included the **new Supplementary Fig. 10** where we show the predicted secondary structure of the tRNA^{Thr} 3' half. In addition we applied ionic conditions to the tRNA half that should either allow or prevent RNA G-quadruplex formation. The presented polyacrylamide

gels do however not show any signs of G-quadruplex formation for this tRNA fragment.

5. *In response to starvation stress, mature tRNA^{Thr} lacks CCA end. Do other tRNAs or their 3'halves also tend to lack CCA ends under such stress? Discuss in the Results*

Our comments:

The referee is indeed correct that 3'-CCA shortening of full length tRNAs under nutritional stress appears to be a more widespread phenomenon in *T. brucei* (and thus similar to human tRNAs; see Czech et al. Plos Gen., 2013; ref. 14 of the revised version). We show these data in the **new Supplementary Fig. 1a** and discuss and refer to these findings on page 6 (Results) and page 14 (Discussion).

6. *Optional: Addition of CCA-end to tRNA-Thr half limits their ability to stimulate translation. Does this suggest that effect is length-dependent or CCA-dependent? How will a variant of tRNA^{Thr} half with any other triplet behave in in vitro translation assay?*

Our comments:

This is an interesting thought. To address this question we have used the tRNA^{Thr} 3' half and have added a 3'-GGU sequence. As can be seen now in the **new Fig. 5a and b**, also this 3' extension prevents the tRNA^{Thr} half to elicit its stimulatory role on translation both *in vitro* and *in vivo*. Thus it seems that the actual length of the tRNA^{Thr} half is crucial for its mechanism of action during protein biosynthesis.

7. *Reference 12 is incomplete, should be : "Heat shock causes a decrease in polysomes and appearance of stress granules in trypanosomes independently of eIF2 α phosphorylation at threonine 169, the residue equivalent to serine 51 in mammalian eIF2 α "*

Our comments:

We have re-checked this citation (now reference 18 of the revised version) and the title is correct as written in the initial submission of this manuscript.

Comments to referee #2:

The data in Figure 1

The sequencing of tRNA to assess their level under different conditions is not a trivial task. This is due to the presence of many modifications that makes the RT fall on the way...this is one of the reason why it is easy to identify 3' half of RNA.

Our comments:

The referee is certainly correct that post-transcriptional modifications can hamper reverse transcription (RT) and thus can lead to premature stops that would flood cDNA libraries and thus RNA-seq experiments. Indeed we were unable to detect full length tRNA sequencing reads in our cDNA library. However, this is not what we did observe with tRNA halves. 3' halves are actually less abundant compared to the 5' counterparts in our tRNA sequencing analyses as well as in our new comprehensive northern blot data set (see **new Supplementary Table 1** and **Supplementary Fig. 1**). We therefore conclude that tRNA modification-dependent RT stops did not significantly skew our data on tRNA-fragment reads.

To overcome this problem and to be able to evaluate changes in the level of tRNA is to use the thermostable group II intron reverse transcriptases (TGIRTs) (see Zheng et al. Nature Methods 2015). Because of the modification on tRNA it is not clear if changes in the very distinct population (30 nt) is due to changes in the level of the RNA or due to changes in modification. The data presented in this Figure seems to be done only once and the actual data (RNA-seq) is not presented (In Supp). These days one need to at least repeat such an experiment three times and provide statistical significance of the data.

Our comments:

We totally agree with the referee that in order to draw solid quantitative conclusions from RNA-seq experiments, at least two biological replicates are needed. However, we would like to point out here that our RNA-Seq experiment of ribosome-associated small ncRNAs (rancRNAs) was intended as a screen to gain insight into which RNA molecules and which RNA classes are ribosome-associated at all. It was never intended to use the sequencing information as quantitative measure for rancRNA expression. Therefore and in accordance to referee #2's valid point, we have responded in the following ways: **(1)** to strengthen the quantitative conclusions of our tRNA halves expression data, we have performed an extensive series of northern blot analyses targeting all tRNA species and tRNA halves in *T. brucei*. These data are now presented in the **new Supplementary Fig 1** and **new Fig. 1d**. This experimental approach allowed us to draw solid quantitative conclusions on (i) the expression levels of tRNA halves under different stress conditions, (ii) the transcriptome-wide 3'-end trimming phenomenon of cytoplasmic tRNAs. We regard the northern blot approach particularly superior to RNA-Seq when it comes to tRNAs and tRNA-derived fragments because we can circumvent two crucial problems, namely the bias due to cDNA library preparation and problems of reverse transcription by modified tRNA nucleosides (as correctly pointed out by reviewer #2). **(2)** we have globally toned down in the text or removed entirely quantitative

conclusions from the bioinformatics analyses (e.g. the previous Fig. 1c was removed completely).

Concerning the criticism that the actual RNA-seq data are not presented:

Our comments:

We probably have not made it clear enough in the initial submission, but **all the RNA-seq data were publicly available** from the day of submission of the first version of this work. To more explicitly highlight this fact we have added in the Results part (page 4) a sentence with the accession number. In addition we have compiled all tRNA-derived sequencing information in a separate Excel file (**new Supplementary Table 1**).

There is a very rich literature about half tRNA in trypanosomatids that is not mentioned in the paper and these studies also provide information regarding change in half tRNAs. This study does not compare their data to others. In many cases, the half tRNAs are also secreted by the cells via different type of vesicles suggesting that these might be transmitting signals to the parasite population or to the host...(for instance Lambertz et al., 2015)

Indeed half tRNA may function to control innate immunity that is very relevant for infectious organisms.

Our comments:

We have indeed not covered the available literature on tRNA fragments in Trypanosomatids adequately in the initial version of the manuscript. We'd like to thank the referee for pointing this out. We have now included a new paragraph in the Discussion (page 12-13) including 5 additional references where we summarize tRNA halves biology in *T. cruzi* and *Leishmania* and also point out the fact that tRNA halves have been found in extracellular particles possibly capable of "communicating" with other parasites or mammalian host cells.

Figure 2: The explanation why the phenomenon of half tRNA formation during starvation and the inhibition during recovery in the two life stages is not convincing. The difference might be coming from the biological relevance for starvation in the two life stages.

Our comments:

In Fig. 2 we show data on tRNA^{Thr} 3' half production both in the procyclic and the bloodstream forms of *T. brucei*. While in the procyclic stage the tRNA^{Thr} 3' half is only produced upon starvation (and during stationary phase) (Figs 2a and b), in the bloodstream form it is present also during exponential growth. When transferring the parasites back to rich media the levels of this half remains constant in both forms. In no case did we see any "inhibition during recovery" as mentioned by the referee. As mentioned above, the halves levels remain constant during recovery and there is in fact no difference in the tRNA half pattern between starved procyclic and bloodstream forms (compare Fig. 2c and d). Only during the exponential growth did we see differences between

the two life stages. In the relevant text parts of the manuscript we merely describe these findings (page 6) and thus we do not intend to “explain a phenomenon”.

Figure 4: The association of half tRNA should be checked by in situ with the tRNA and DHH1 (IFA or fluorescence). There are studies in trypanosomes showing that half tRNA are found in stress granules (Gracia-Silva Mol. Biochem. Parasitol 2010).

Our comments:

In our submission we provide evidence that the tRNA^{Thr} 3' half binds to ribosomes and can stimulate protein biosynthesis *in vivo* during stress recovery and during *in vitro* translation. In order to understand the mechanism of action behind these observations we were interested in investigating whether stress granule (SG) formation/disassembly is affected by this tRNA half. It has been reported before that mammalian tRNA halves inhibit translation and stimulate SG assembly (P. Anderson & P. Ivanov labs). In *T. cruzi* Garcia-Silva et al., (2010) have detected tRNA halves in cytoplasmic granules whose “structure and composition.....is unknown” (quote from Garcia-Silva et al., 2010).

Since changes in SG formation/resolution might potentially explain the observed effects on *in vivo* protein synthesis we decided to look whether or not the tRNA half under investigation affects the dynamics of SG. As we show in Supplementary Fig. 4 of the revised manuscript, the presence of elevated amounts of tRNA half does not change SG formation/resolution. This, we think, is an important piece of information to develop the scientific story of our submission, since our data do not support a SG contribution. Therefore, since we do not see any effects on SG dynamics, we are not convinced that performing *in situs* to check if the tRNA^{Thr} 3' half is part of genuine SG would add to the mechanism of action. We feel that this would be beyond the scope of this manuscript where we focus on the role of this ribosome-bound tRNA half on translation. We did, however, add a sentence in the Discussion (page 16) where we state that we cannot exclude the possibility that the tRNA^{Thr} 3' half has also roles beyond translation control.

Figure 5: tRNAs are heavily modified it is strange that in vitro transcribed RNA has a biological role... the CCA at the 3' end may also protect the half tRNA from degradation...

Our comments:

We like to point out here that the tRNA^{Thr} 3' half only shows its stimulatory effect on translation *in vivo* and *in vitro* when the 3' CCA is absent. Therefore a protective role of the CCA on the tRNA half function is not an issue. Furthermore we have evidence that electroporated tRNA^{Thr} 3' halves lacking the 3' CCA are stable in the cell for several hours (see Supplementary Fig 7).

The biological role of post-transcriptional modifications was and still is in the focus of dedicated research for decades, yet the function of many of them still remains enigmatic (e.g. the roles of many rRNA or tRNA modifications, even though often conserved, remain to be uncovered). Thus we are less surprised that the tRNA^{Thr} 3' half does possess activity in the absence of any modification. There are ample of examples in the literature demonstrating that *in vitro* transcribed or chemically

synthesized RNA molecules possess a biological role (e.g. 23S rRNA can be reconstituted into functional ribosomes, sgRNA or siRNA function in knock-out and knock-down approaches *in vivo*, respectively, tRNAs *in vitro* transcripts can be aminoacylated and are active as substrates for protein synthesis, ect.).

Figure 6: The fact that in vivo selected half-tRNA has only a very minor stimulatory effect is a bit confusing and bothering. Is it possible to quantitate the amount of RNA used in transfection? I think that the main difference between in vivo and in vitro half tRNA is not the amount but the fact that the tRNA are either modified or not...

Our comments:

We followed the referee's suggestion and have repeated the affinity purification of endogenous tRNA^{Thr} 3' halves and have analyzed the samples alongside known amounts of *in vitro* transcribed halves by northern blot analysis (**new Supplementary Fig. 9**). This approach allowed us to calculate the actual amount of RNA used for *in vitro* translation (not "transfection" as stated by the referee). The data shown in the new Supplementary Fig. 9 reveal that only 1 pmol affinity purified halves could be used in the experiments shown in Fig. 6c. This explains, as we think in a straight forward manner, why the observed effects in Fig. 6c are so small.

The referee's comment that "*the main difference between in vivo and in vitro half tRNA is not the amount but the fact that the tRNA are either modified or not*" is a valid one, yet not supported by our experimental data.

In sum, the phenomenon is interesting, but as I outlined above a few of the experiments need to be repeated but I think that to publish the study in Nature communication one needs to show the mechanism of the activation

Our comments:

We thank the referee for this encouraging comment and we agree that mechanistic insight on how such a small tRNA half molecule can boost translation of ribosomes during stress recovery in *T. brucei* would strengthen this study. Motivated by these comments we performed the following experiment: we assembled *T. brucei in vitro* translation reactions in the presence or absence of the tRNA^{Thr} 3' half. The reactions were then stopped by cycloheximide addition at different time points. Subsequently the ribosomes were harvested by passing them through a sucrose cushion and the amount of ribosome-bound mRNA was assessed by northern blotting. These **new data** are shown in **Fig. 5d** and demonstrate that significantly more tubulin mRNA is ribosome-associated in the presence of the tRNA halves. This suggests that translation initiation is most likely the phase where this tRNA half influences protein biosynthesis. We think these new findings provide promising mechanistic insight into the mode of action of the tRNA^{Thr} 3' half. We discuss this scenario in more detail in the Discussion part (page 14) of the revised manuscript.

Reviewers' Comments:

Reviewer #1:

Remarks to the Author:

The revised version of the manuscript by Fricker et al. further strengthen initial conclusions made by authors in the original submission. The work is sound, novel, has broad interest for RNA biology community. All statistical analysis is adequate, sequencing data is publically available and methodology is described in great details.

This work provides insights into biology of *Trypanosoma brucei* connected to the biogenesis of novel ncRNAs derived from tRNAs and also nicely describes new mode of physical and functional interactions between tRNA fragments and translation machinery, especially in stimulation of translation. Of note, this is the first paper that focuses on the functions of 3' tRNA halves, the tRNA fragments that are largely unexplored (when compared to multiple studies on 5' tRNA halves).

All my requests were addressed adequately (results described in Sup. figures 1A-B, 5, 8b, 10) and I am writing in strong support for its publication in Nature Communications.

Reviewer #2:

Remarks to the Author:

The revised manuscript has much improved and especially the attempt to find how the half tRNA stimulate translation. The manuscript is also improved by giving more information and comparing to what we know about half tRNA and their secretion by exosomes, as well as on the function of half tRNA in other eukaryotes including the way these are processed.

All the criticism was addressed at different levels. Additional experiments were performed and the data was refined. I am still not happy with presenting a single RNA-seq experiment. The authors tried to verify what they think is important by Northern analysis and it is not clear to check if in all cases a good correlation is found with RNA-seq and the Northern.

The stimulatory effect one observes is there...not so big but consistent and time will tell of others will see it and if we finally understand how this works...

I still think that the modification of tRNA is important...and the comment regarding the role of modifications is very annoying. Great progress was made in the RNA modification field. This is one of the most studied field in RNA Biology these days. The fact that so many modification exist on tRNA tells us that it is important...Yes, it is complicated to study and very sensitive assays are still missing...but it is important and is important for the function of these molecules even if we still do not know always why...

As for the absence of CCA in the tRNA fragment. It is possible that upon starvation CCA is not added...this is step in maturation of tRNA and might be stopped under harsh conditions.

Minor comments

"The fact that the amount of halves from individual tRNAs changes as

. function of stress application, argues against post-transcriptional modifications significantly influencing cDNA library construction under the applied experimental settings"

I think this sentence needs to be removed. The modification themselves can be changed under the harsh cues imposed here such as starvation.

In the methods section: 24 µl sample mix was composed of 3 µl DMSO, 7 µl *T. brucei* cell extract containing 20 mM RVC (120 OD260/ml), 7.5 µl ribosome buffer B etc...How much extract was used or rather from how many cells was it made...this is important. Has the authors used a protocol devised by others if so a reference should be added. If not more details should be given.

Point-by-point response:

Comments to referee #1:

The revised version of the manuscript by Fricker et al. further strengthen initial conclusions made by authors in the original submission. The work is sound, novel, has broad interest for RNA biology community. All statistical analysis is adequate, sequencing data is publically available and methodology is described in great details.

*This work provides insights into biology of *Trypanosoma brucei* connected to the biogenesis of novel ncRNAs derived from tRNAs and also nicely describes new mode of physical and functional interactions between tRNA fragments and translation machinery, especially in stimulation of translation. Of note, this is the first paper that focuses on the functions of 3' tRNA halves, the tRNA fragments that are largely unexplored (when compared to multiple studies on 5' tRNA halves).*

All my requests were addressed adequately (results described in Sup. figures 1A-B, 5, 8b, 10) and I am writing in strong support for its publication in Nature Communications.

Our comments: We thank this referee for his/her constructive feedback during this review process.

Comments to referee #2:

The revised manuscript has much improved and especially the attempt to find how the half tRNA stimulate translation. The manuscript is also improved by giving more information and comparing to what we know about half tRNA and their secretion by exosomes, as well as on the function of half tRNA in other eukaryotes including the way these are processed.

1.) All the criticism was addressed at different levels. Additional experiments were performed and the data was refined. I am still not happy with presenting a single RNA-seq experiment. The authors tried to verify what they think is important by Northern analysis and it is not clear to check if in all cases a good correlation is found with RNA-seq and the Northern.

Our comments: As already pointed out in our previous rebuttal letter, the RNA-Seq experiment on ribosome-associated ncRNA (rancRNAs) was solely performed as screen to gain insight into which RNA species are present on *T. brucei* ribosomes. Subsequently we decided to focus on tRNA-derived fragments and in particular on one 3'-half from tRNA^{Thr}. To gain quantitative insight into the abundance of tRNA-derived fragments in *T. brucei* cells we now provide northern blot analyses performed on **three biological replicates** targeting all tRNA species under different growth/stress conditions. These blots and the quantifications are shown in the **new Supplementary Figure 1**. These northern blot data show that most significant tRNA fragmentation occurs during conditions of nutrient deprivation (during stationary phase and after incubating the cells in PBS). A very similar trend was already obvious after the RNA-Seq analyses, however we regard the northern blot approach as a superior method to assess tRNA abundance (due to the already mentioned

reverse transcription hurdles that originate from tRNA modifications which can affect cDNA library construction).

2.) *I still think that the modification of tRNA is important...and the comment regarding the role of modifications is very annoying. Great progress was made in the RNA modification field. This is one of the most studied field in RNA Biology these days. The fact that so many modification exist on tRNA tells us that it is important...Yes, it is complicated to study and very sensitive assays are still missing...but it is important and is important for the function of these molecules even if we sill do not know always why...*

Our comments: The biological role of tRNA modifications (or RNA modifications in general) is indeed a hot topic today. As referee #2 pointed out correctly, sensitive assays to elucidate the roles of RNA modifications are mostly lacking. Therefore it is most of the time impossible to deduce the role of an RNA modification based on its mere presence. However, whenever a synthetic RNA that lacks modifications is biologically active (as it is in our case with the tRNA^{Thr} 3' half) one can exclude that a modification is essential for this particular function. This does by no means imply that tRNA modifications are not important, just not for the observed effect we report in this manuscript. To further address and seal this issue experimentally, we have **analyzed the tRNA modifications** of affinity purified endogenous tRNA^{Thr} during exponential growth and after starvation by **LC-MS/MS**. The MS data demonstrate that the tRNA^{Thr} modification pattern does not change as a function of stress application. We regard this as direct evidence that the production of the tRNA^{Thr} 3' half and most certainly also the physiological role of this fragment does not depend on (altered) modifications. We show the MS data in the **new Supplementary Fig. 8** and we devote a new paragraph in the *Results* chapter of the manuscript (page 10) to post-transcriptional modifications and their roles for RNA biology.

3.) *As for the absence of CCA in the tRNA fragment. It is possible that upon starvation CCA is not added...this is step in maturation of tRNA and might be stopped under harsh conditions.*

Our comments: In this new point, referee #2 suspects that the reason for the observed lack of the 3'-CCA on the tRNA^{Thr} 3' half is due to the fact that full length tRNA^{Thr} might have never received a 3'-CCA tail during starvation conditions. Our data however demonstrate that during PBS starvation, conditions when new tRNA transcription is negligible, essentially all endogenous tRNA species get trimmed by losing their 3' CCA (**Supplementary Fig. 1a** & manuscript in preparation). Thus this indicates that tRNA 3' halves in *T. brucei* are processed from full-length tRNAs whose 3'-CCA got removed during starvation by a so far unknown RNase. We discuss these data in the *Results* on page 6.

Minor comments

"The fact that the amount of halves from individual tRNAs changes as function of stress application, argues against post-transcriptional modifications significantly influencing cDNA library construction under the applied experimental settings"

I think this sentence needs to be removed. The modification themselves can be changed under the harsh cues imposed here such as starvation.

Our comments: Even though we show in this revised version that the modification pattern of tRNA^{Thr} does not change during starvation (Supplementary Fig. 8), we cannot extrapolate to all *T. brucei* tRNAs. Therefore we followed the suggestion of the referee and have removed this sentence.

In the methods section: 24 µl sample mix was composed of 3 µl DMSO, 7 µl T. brucei cell extract containing 20 mM RVC (120 OD260/ml), 7.5 µl ribosome buffer B etc...How much extract was used or rather from how many cells was it made...this is important. Has the authors used a protocol devised by others if so a reference should be added. If not more details should be given.

Our comments: In the *Methods* section we clarify, how the *T. brucei* cell extract used for *in vitro* translation was prepared. Furthermore we also give details how many cells were actually used to prepare the respective *in vitro* translation extract.